# Reversible On-Off Photoswitching of DNA Replication Using a Dumbbell Oligodeoxynucleotide

**DOI:** 10.3390/molecules27248992

**Published:** 2022-12-16

**Authors:** Yu Wang, Heming Ji, Jian Ma, Hang Luo, Yujian He, Xinjing Tang, Li Wu

**Affiliations:** 1School of Chemical Sciences, University of Chinese Academy of Sciences, Beijing 100049, China; 2Zhejiang Institute of Mechanical and Electrical Technician, Yiwu 322000, China; 3State Key Laboratory of Natural and Biomimetic Drugs, School of Pharmaceutical Sciences, Peking University, Beijing 100191, China

**Keywords:** DNA extension, azODNs, UV, control

## Abstract

In most organisms, DNA extension is highly regulated; however, most studies have focused on controlling the initiation of replication, and few have been done to control the regulation of DNA extension. In this study, we adopted a new strategy for azODNs to regulate DNA extension, which is based on azobenzene oligonucleotide chimeras regulated by substrate binding affinity, and the conformation of the chimera can be regulated by a light source with a light wavelength of 365 nm. The results showed that the primer was extended with Taq DNA polymerase after visible light treatment, and DNA extension could be effectively hindered with UV light treatment. We also verify the reversibility of the photoregulation of primer extension through photoswitching of dumbbell asODNs by alternate irradiation with UV and visible light. Our method has the advantages of fast and simple, green response and reversible operations, providing a new strategy for regulating gene replication.

## 1. Introduction

Light is particularly attractive for the control of biological functions because it provides highly accurate spatiotemporal resolution as an external trigger that does not lead to sample contamination [1,2,3,4,5,6]. Azobenzene has been shown to be a reversible manipulated light-response molecule [7]. It is possible to control biological functions by specific wavelengths of light by the incorporation of azobenzene units into nucleic acids in proper structural positions [8]. Such molecular photoswitches undergo a light-induced reversible change in their structure that results in a change in their properties, for example, geometry, polarity, flexibility or degree of conjugation, etc. These effects may be translated into changes in the functioning of biological systems, as already presented by the photocontrol of aptamer recognition [9], RNA digestion [10], enzyme activity [11] and gene expression [12].

Among the many operational strategies for biological function, azobenzene is an excellent choice because the photo-induced reversible trans-cis isomerization of azobenzene is accompanied by geometry, with large changes in considerable influence on the structure and activity of biomolecules [13]. Azobenzene can be introduced into DNA, using the anticipant structure of azobenzene to regulate DNA ligature or departure [14]. Azobenzene is affected by ultraviolet light, its structure changes from trans to cis, irradiation with visible light will change from cis to trans, and this reversible isomerization will affect the structure of DNA double strands. Based on this technique, Michael et al. designed two different types of azobenzene derivatives, actinic DNA walkers under the control of different light wavelengths [15]. A machine-like DNA enzyme that digests RNA by photomodulating the topology of the enzyme’s active site has recently been reported [16]. Modulation of substrate binding affinity is a simpler, more direct and easier-to-perform strategy for manipulating enzyme activity than modulation of the topology of the active site.

The pioneered application of azobenzene molecules as optical switches in nucleoside surrogates was in the photomodulation of DNA primer extension by DNA polymerase, and the modularity method is demonstrated by oligonucleotides bearing azobenzenes. Azobenzene is introduced into the phosphate backbone of the antisense oligodeoxynucleotides close to 5′, and this chain can form a stable duplex with the template when azobenzene is a trans form, forming an oligomer chain as a blocking unit, and the primer extension stops at the position of 5′-azobenzene. For the cis form, the stability of the oligomers is greatly reduced, exposing the template single strand, and primer extension can reach the end of the template [17]. Azobenzene switches also quickly undergo cis- and trans- isomerization due to the influence of temperature. However, when trans-azobenzene is inserted into a base in the DNA duplex, the photoisomerization efficiency of trans- to cis-form is generally not high, and the method has so far been limited to short-modified oligonucleotides, which may have limited effects under physiological conditions [18]. As a prominent example, tethering of azobenzene to DNA can be used for the photomodulation of RNase H assays [19].

So far, the application of an optical switch as a nucleoside substitute in biological environments is relatively limited. This could be due to several factors. First of all, most optical switches are isomerized under ultraviolet light, which is harmful to organisms. In addition, photoisomerization efficiency is generally poor under physiological conditions. Finally, the free volume available for isomerization is limited at temperatures below oligonucleotide T_m_ (melting temperature), resulting in low photoisomerization efficiency under irradiation. Including any non-natural skeleton joints in long stranded RNA or DNA can be challenging because modifications can prevent primer extension of DNA polymerase. Therefore, the possibility of including azobenzene-modified nucleotides in genes by PCR is limited [20,21,22]. Molecular photoswitches as nucleoside substitutes is a design that has important implications for future in vivo applications, initially in a closed state with antisense oligodeoxynucleotides (asODNs), which can silence disease genes caused by abnormal gene expression or normal gene overexpression.

We recently proposed the use of a dumbbell structure with azobenzene DNA as modulators. Given our previous research, two hairpins at two terminals of an antisense oligodeoxynucleotides (asODNs) form a dumbbell structure with azobenzene as hairpin loops (azODNs), which successfully controlled RNA digestion by reversibly photomodulating the hybridization of asODNs to target RNA molecules [23]. Here, we hope to apply azODNs to regulate primer extensions by manipulating asODNs binding to template molecules artificially through the trans- and cis-structures of azobenzene. This method is very flexible and can synthesize fragment-length azODNs in vitro for the target gene sequence you want to study and even develop it for use on plasmids in cells. This will lay the foundation for exploring the mechanisms of action between substrates in the presence of polymerase.

In this study, according to the commonly used phosphoramide monomer synthesis method (Appendix A), 2-Cyanoethyl-4-O-{[4-(4,4′-dimethoxytrityl)-O-methyl-diazenyl)]benzyl}-N,N′-diisopropylaminophosphoramidite is introduced into asODNs by using a DNA synthesizer. asODNs connected to azobenzene can be used to achieve primer extension on-off optical switches (Figure 1). Herein, azODNs are designed to be 16 nt (Az1, Az2 and Az3), 18 nt (Az4, Az5 and Az6) and 20 nt (Az7, Az8 and Az9) long asODNs, with two azobenzene linked hairpin structures, and each hairpin contains inhibitory sensory chains of 4-mer (Az1, Az4 and Az5), 5-mer (Az2, Az5 and Az8) and 6-mer (Az3, Az6 and Az9) lengths at both ends of these ODNs (Table 1). Then, photomodulation of the primer extension by Taq DNA polymerase are evaluated in gel shift assay. Thus, the azobenzene-based photoresponse ODNs can convert the light signal directly into DNA systems and this signal can control genetic information.

## 2. Results

### 2.1. Photoisomerization of Azobenzene-Modified DNAs (Hairpin asODNs and azODNs)

Previous work in this research group has known the light time required for cis- and trans-isomerization of azobenzene. After attaching azobenzene into an oligonucleotide single strand, the illumination time is controlled for cis- and trans-isomerization of azODNs. Azobenzene-modified DNA (AH1 and Az5) is dissolved in 1 × PBS. The solution is annealed at 95 °C to form a stable structure. Trans is a thermally stable form of azobenzene with a distinct shoulder peak in the ultraviolet spectrum at 335 nm (π − π*). For AH1 solution, after ultraviolet illumination at 365 nm, it can be clearly observed that the peak intensity at 335 nm decreases with increasing illumination time (1 min, 2 min, 3 min, 4 min and 5 min), with the largest decrease being in the first 3 min. As UV illumination continues, the absorption value is still decreasing, but the decrease is slow (Figure 2a). Similarly, the dumbbell-type nucleic acid, Az5, also has this phenomenon (Figure 2b). AH2, AH3, Az8 and Az5 (Appendix A) solutions showed similar changes at 335 nm after UV irradiation. They all have rapid photosensitivity and can change azobenzene from trans to cis under brief light. After half an hour of illumination, it was immediately detected that the absorption value was almost undetectable at 335 nm (Appendix A). After a long period of ultraviolet light, the peak of the characteristic absorption peak of nucleic acids at 260 nm becomes higher, indicating that the DNA has changed from a double strand to a single strand during the illumination, and we will use the thermodynamic stability value for further verification in the next step.

### 2.2. Thermodynamic Stability of azODNs

The alteration of thermodynamic stability caused by isomerization of azobenzene is a necessary prerequisite for azODNs to control DNA elongation. Naturally structured hairpin nucleic acids A1, A2 and A3 were compared with azobenzene hairpin nucleic acids AH1, AH2 and AH3. The melting curves of sequences containing different azobenzene derivatives varied significantly, and the melting temperature for each sequence was calculated from the melting curves (Table 2 and Appendix A). The results showed that the T_m_ values of the azobenzene-modified oligonucleotides changed significantly before and after illumination, with trans AH1, AH2 and AH3 being 53 °C, 58.6 °C and 53.6 °C, respectively, and the T_m_ values of the cis-structure being 38.9 °C, 40 °C and 40.4 °C, respectively, and their differences could reach up to 18.6 °C. This phenomenon may be due to the cis- and trans-structure of azobenzene affecting the melting of the double strand; trans-azobenzene helps the formation of hairpin structure, and the cis-azobenzene derivative blocks adjacent complementary matching base pairs. Compared with natural nucleotides, azobenzene-modified hairpin oligonucleotides have higher T_m_ values, and the difference ranges from 18.3 °C to 25.2 °C. This phenomenon is more pronounced for azODNs (Az1-Az9), where the introduction of two azobenzenes leads to the higher thermal stability of nucleic acid strands. Significantly, there was a decrease in T_m_ values in the range of 9.9–22.9 °C from the trans form and the cis form for all azobenzene-linked dumbbell oligonucleotides (Table 2, Appendix A).

### 2.3. Analysis of Primer-Extension Reactions with azODNs

We first attempt to construct a photoresponsive polymerase reaction system. The Primer14 is labeled a fluorescein moiety (FAM) at its 5′-terminus, and a Tem32 DNA template is used for the DNA replication in vitro. A 1:2 ratio of azODNs to DNA template was chosen for the photoregulation of primer extension in the presence of 1 U Taq polymerase for different times at 37 °C. Under irradiation with UV light (365 nm, 30 min), the DNA elongation was blocked by azODNs integrated with the template. However, it will produce a full-length product due to less firm binding between azODNs and the template after irradiation with visible light. Apparently, the extension products were thus visualized and quantified using denaturing polyacrylamide gel electrophoresis (PAGE) (Figure 3 and Figure 4). The azODNs may have multiple bands on the gel due to the presence of both cis- and trans-structures (Appendix A). Normally, primers extend to the end until obstructed by azODNs. Considering the different number of antisense oligonucleotides complementarily paired with the template strand, the base lengths of the two hairpin-protected strands of dumbbell-type nucleic acids are inversely proportional to the length of the complementarily paired antisense oligonucleic acid strand, and multiple short chain products were obtained except full-length products.

We can find that the full-length DNA obtained from one template rely on whether the light is used or not. Before UV irradiation, azODNs mostly take the trans form by annealing the reaction solution and cannot be stabilized by binding to the template. It is competitively excluded from the template by polymerase, and the DNA polymerase moves along the template in the 3′–5′ direction during the polymerization. Clearly, relatively rapid dNTP incorporation into the primer directed by a trans-modulator was observed, and the percentage of the full-length product of the primer extension reactions, defined as PCR%, are 12.0% for Az1, 29.2% for Az2 and 33.3% for Az3 (Figure 4a), respectively. In comparison with Az1, the azODNs Az2 and Az3 have the same 16-nt-long complementary asODN sequence but more stable dumbbell structures with two and four more base pairs in two binding arms, which led to only 6- and 4-nt-long complementary asODN segments left for target template binding, respectively. So, Az3 has a higher elongation than Az1 and Az2. A remarkable decrease in the efficiency primer extension with exposure to UV light was observed after 20 min. The primer extension greatly reduced 2.4% for Az1, which was about a 5.0-fold decrease. To decrease DNA replication for cis-azobenzene-linked azODNs to a larger extent, then, azobenzene-linked azODN with 18 and 20 nt antisense oligodeoxynucleotides fragments complementary to the template is used to assess the photomodulation of primer extension. The result showed much better photomodulation efficiency (9-fold) from trans-Az8 (13.5%) to cis-Az8 (1.5%), resulting in the synthesis of little full-length products after UV light. The primer extension greatly decreased from 9.0% to 1.5% for Az4 and from 20.0% to 3.2% for Az9, despite being the different-lengths (18 nt, 20 nt) complementary azODNs, which were about a 7.5-fold and 6.3-fold decrease, respectively. For further increase of the reaction time to 30 min (Figure 4b), Az1, Az7, Az8 and Az9 also displayed an obvious lag in incorporating dNTPs directed by a cis-modulator relative to a trans-modulator; the modulation efficiency was a 4.2-fold decrease at least upon UV light irradiation. Because of UV irradiation, the two binding arms of the asODN are opened and stably bound to the template under the synergistic effect of free nucleotides, and the DNA elongation terminates at the 5′ end of modulator. Primer extension for Az8 with two 5-nucleotide hairpin stems was a 6.3-fold decrease from 37.5% to 5.9%, while for Az9, a low-level difference of primer extension was observed before (50.8%) or after (20.0%) UV irradiation. Unsurprisingly, the control efficiency of most experimental groups decreased gradually with the action of polymerase and the limitations of the controller itself (Figure 4c).

DNA polymerase requires single-stranded DNA as a template for primer extension, and the other polymerase work tends to unwind between the controller and the template. We designed and synthesized Tem32 and Tem42 templates of different lengths (32 mer and 42 mer), and quantitative analyses of the running start assays were performed by the efficiency of the extension as a function of enough time (90 min) (Figure 5). The data showed that although the efficiency of the enzyme recognition by the controller returned to normal, there were still differences in the elongation of the system primers, and the elongation of the control group after UV light exposure was still not significant. The performance of azODNs structures with four base pairs in two binding arms and Az8 in the PAGE is visually distinguishable. Because the controller itself is more closely paired with the template, these results showed much better photomodulation efficiency (2-fold) from trans to cis.

### 2.4. Elongation Kinetics of Primer Extension by Cis- and Trans-azODNs

In the case of dumbbell-type nucleic acid Az9 binding to Tem32, the efficiency of primer extension is evaluated by the control of illumination using UV external illumination for up to 60 min (Figure 6a). The elongation of primers in Az9 is greatly reduced after UV irradiation, indicating that cis-Az9 forms a partial double strand with the template chain, which hinders the extension of the enzyme. Compared to non-UV treatment, the elongation efficiency shows a significant lag, with the elongation efficiency almost 4.3 times that of UV treatment (Figure 6b). Although the T_m_ value of the cis- and trans-structure is greater than 37 °C, there is slow cis- and trans-isomer conversion at 37 °C. So, with the wireless extension of time, the same conversion effect will eventually be achieved, but we can control the extension effect within a certain time range, and we can also achieve the purpose of regulating the extension.

### 2.5. Reversible Photomodulation of Primer Extension by azODNs

Next, the effect of azODNs on reversible photomodulation of primer extension was investigated. Alternating light irradiation between ultraviolet and visible light was used to switch the primer extension (Figure 7). In the Analysis of primer extension reactions with azODNs, it was found that Az8 and Az9 had better primer extension regulation results at 20 min and 40 min, and Az8 and Az9 had the same complementary bases for the template chain, and different hairpin base pairs at both ends; so, Az8 and Az9 were selected for controllable primer extension experiments.

The reaction was started in a dark condition and then incubated at 37 °C for 12 min. The sample was then irradiated with UV light (2 min), and DNA primer extension was reduced and further incubated at the same condition for another 12 min. In the end, visible light irradiation was carried out again, and extension was efficiently induced. Aliquots of the solution were taken out every 6 min except for irradiation time during the entire experimental process. We can clearly find out when UV irradiation and the slope of linear fitting data were 0.458 and 2.48 for Az8 and Az9, respectively. With isomerization to the cis from trans form following 2 min of UV irradiation, primer extension slowed down significantly, and linear fitting of the percentages of primer extension was observed with the slope of only 0.208 (Az8) and 0.65 (Az9). The initial primer extension was partially recovered following further visible light irradiation, and the slope (1.35 for Az8, 1.81 for Az9) of linear-fitting data also obviously increased. This observation of light-induced primer extension indicated that inhibition and activation of primer extension may be reversibly modulated through photoswitching of dumbbell asODNs by alternate irradiation with UV and visible light.

## 3. Discussion and Conclusions

We have synthesized and evaluated azo-containing hairpin nucleic acids and azo-containing azODNs, a new class of asODNs that can be photochemically controlled. We have shown that asODNs can be inactivated in an efficient manner to minimally inhibit primer extension by using UV light at 365 nm, and that reactivation by visible light can release asODNs to recover the polymerization reactions. We consider this to be an improvement over the traditional cage approach [24]. Azobenzene has the potential to activate or close several times during treatment, whereas the traditional light cage releases its full load once activated and is not affected by light afterward. For example, the photocage group needs to be synthesized with the base of the nucleotide monomer. Once the NPOM is left by ultraviolet light, it cannot be restored to the state of modified nucleotide monomer, and NPOM is a reaction product after illumination, which may also have a side effect on organisms. In addition to the advantage that azobenzene can be switched on and off repeatedly, chemical synthesis of azobenzene is easier than the traditional photocage group NPOM. It has the added effect of not producing potentially serious by-products, such as reactive oxygen species (ROS), by removing photocage groups. Azobenzene has its own challenges. Thermal relaxation back to the more stable trans conformation limits its use as a long-term therapy (beyond two half-lives) [25,26]. The azobenzene moiety itself is also metabolized by thiol reduction or the metabolism of glutathione, which is present in millimolar quantities in cells and, therefore, its future effectiveness as a reversible tool needs to be fully investigated [27]. In addition to UV illumination control, chemical groups can also be modified on azobenzene to introduce other control light sources, such as red light [28] and green light [29], which further expands the illumination sources and avoids the selection of light sources harmful to nucleic acids and organisms. These modifications also allow us to better control the time and space of asODNs and can be applied to time-sensitive dose control.

Other possible methods include incorporating multiple azobenzenes into the positive strand or even modifying the antisense strand. However, prolonged UV irradiation would lead to side effects, such as cell toxicity. For azobenzene-linked hairpin nucleic acids and dumbbell nucleic acids, the controllable irradiation time was required to generate the fully active molecule. In the isomerization study of azobenzene nucleic acid chains, samples were treated with 365 nm UV light, and by measuring the absorption value at 335 nm, the results showed that the change was very slow after 5 min, which was observed by oligonucleotides with azo. When determining the stability of asODNs, the T_m_ value of azobenzene nucleotides is significantly higher than that of natural nucleotides. The 4, 5 and 6 base protective chains linked by 4,4′-dihydroxymethylazobenzene coupling with lengths of 16 mer, 18 mer and 20 mer of azODNs were screened to adjust primer extension efficiency under UV illumination. These azODNs generally have a good photoregulation effect on elongation, especially for Az8 which has up to 9-fold photomodulation efficiency. We also achieved photoregulation in the reversibility of Az8 and Az9 by alternate irradiation with UV and visible light and concluded that the main effect of azobenzene modified DNA strand on the photocontrol behavior of primer elongation is because the enzyme inhibits elongation through the obstruction of template-strand formation during derivation.

In conclusion, we report here a new light-controlled extension technique because we can control primer extension reversibly with light. This new technology may provide the ability to reversibly control the activity of endogenous targets and will aid in the replication of disease genes. Another potential application of azODNs is their use as a biomolecular tool to detect the effects of gene replication in complex and/or related genes in real time, allowing for the smooth extension of several different genes controlled in real time by light. The photoactive modification of azobenzene on the template sequences involved in life activities can be used as a scientific tool in the future and guide the application of nucleic acid drugs. Our future work includes modifying azobenzene to improve biosecurity by using non-UV light to affect switching activity and using multiple light sources to develop logic and gate gene expression control.

## 4. Materials and Methods

**Compound 1.** 4,4′-bis(hydroxyethyl)−azobenzene: 6.0 g (26 mmol) 4−nitrobenzyl alcohol was added to 70 mL NaOH (5.7 M) aqueous solution, and 7.0 g (100 mmol) Zn power was added slowly. After addition of the materials, the solution was refluxed by stirring vigorously. The reaction mixture was filtered after one day, and the filtered solid was suspended in hot methanol many times until the azo components were dissolved completely. Air was bubbled into the azobenzene methanol solution and refluxed for 10 h. Upon concentration on vacuum evaporator, the left methanol solution was slowly cooled to give orange solid 3.2 g, with yield 50%. ^1^H NMR (DMSO, 400 M): δ 7.87 (d, 4H), 7.54 (d, 4H), 5.39 (t, 2H), 4.61 (d, 4H). 13C NMR (DMSO, 101 M): δ 105.84, 146.24, 127.09, 122.34, 62.43.

**Compound 2.** 4-hydroxymethyl-4′-O-(4,4′-dimethoxytrityl)-azobenzene 1.5 g (6.5 mmol) of 4,4′-bis(hydroxyethyl)-azobenzene was dissolved in 30 mL dry pyridine, and some pyridine was distilled to remove water residue in the solution. After cooling the solution, 2.1 g (6.5 mmol) 4,4′-dimethoxytrityl chloride was added with stirring vigorously. The reaction was monitored by TLC. The reaction mixture was concentrated and the residue was purified by silica gel chromatography with CH_2_Cl_2_/MeOH (volume ratio: 100/2) to give orange solids (2.0 g), yield 34%. ^1^H NMR (CDCl_3_, 400 M): δ 7.90 (m, 4H), 7.52 (m, 6H), 7.42 (m, 4H), 7.24 (m, 3H), 6.85 (m, 4H), 4.73 (s, 2H), 4.26 (s, 2H) 3.78 (s, 6H). ^13^C NMR (CDCl3, 101 M): δ 158.56, 152.05, 151.84, 144.99, 144.44, 142.58, 136.19, 130.1, 128.2, 127.93, 127.46, 127.34, 126.87, 122.98, 122.82, 113.22, 86.63, 65.31, 64.40, 55.24.

**Compound 3.** 2-Cyanoethyl-4-O-{[4-(4,4′-dimethoxytrityl)-O-methyl-diazenyl)]benzyl}-N,N′-diisopropylaminophosphoramidite to the solution of 4-hydroxymethyl-4′-O-(4,4′-dimethoxytrityl)-azobenzene (0.160 g, 0.29 mmol) dissolved in 2 mL dry acetonitrile, 0.120 g 2-cyanoethyl-N,N,N,N -tetraisopropylphosphora-diamidite and 21 mg (0.30 mmol) 1H-tetrazole were added. The mixture was stirred under N_2_, and the reaction was monitored by TLC (yield was estimated above 90%). After an hour, the solution was filtered using hydrophobic membrane filter and moved to a vial for application to DNA synthesis without purification, and completion of the reaction was identified by ^31^PNMR. ^31^PNMR (CDCl3, 162 M): δ 148.92, 148.77.

**The synthesis of azobenzene modified DNA.** The balance weighs a 35mg (1 μmol) CPG solid-phase support into a Universal-CPG synthetic column. Edit the oligonucleotide sequence of interest from 5′ to 3′ (from 3′ to 5′ during synthesis) depending on where the different phosphoramite monomers are loaded on the solid-phase synthesizer. All nucleic acid sequences are connected to the solid-phase CPG from the 3′ end to the 5′ end using the DNA/RNA solid-phase synthesizer by removing DMT, ETT activation, coupling, capping and oxidation steps according to conventional synthesis methods. The coupling time of ordinary DNA monomers is 120 s, and the coupling time of azobenzene derivative monomers is increased to 600 s. Until all phosphoramite monomers are successively coupled to the oligonucleotides of CPG, the solid phase of the target sequence is obtained. The average single-step yield is above 90%.

**Purification of the azobenzene modified DNA.** (1) Cut the solid phase. The synthesized oligonucleotides chain is connected to the CPG in a covalent bonding manner, transfer the solid phase CPG to a 2 mL centrifuge tube, add 1 mL of concentrated ammonia, hydrolyze at 50 °C for 8 h, remove the oligonucleotide from the solid phase and deprotect the group, then centrifuge with a centrifuge (2000 rpm), take the supernatant and concentrate it with a DNA concentrator until the ammonia water is completely dry (yellow solids can be seen at the bottom of the EP tube), add 200 μL of deionized water, and filtrate by membrane; (2) separation and purification. The crude DNA in this experiment was purified by Agilent reversed-phase C18 column (5 μm particle size, 9.6 mm × 250 mm) under liquid chromatography conditions: UV detection wavelength 260 nm and 335 nm dual-wavelength detection. Fluidity A is 50 mM TEAA, B is acetonitrile; gradient elution procedure: 0–10 min, 5–40% B; 40–45 min, 40–100% B; 45–50 min, 100% B; 50–55 min; 100–0% B; 55–60 min, 0% B; flow rate 1.0 mL/min; column temperature: 30 °C. Collect the target product according to the peak time of the absorbance at 260 nm and 335 nm of the UV detector.

**Photoisomerization of DNA**. Dissolve all sequences in 1× PBS to prepare a solution, heat in a metal bath at 95 °C for 2 min, and cool naturally to room temperature. The sample solution was processed for other operations then transferred to a quartz cuvette and irradiated with a UV lamp (365 nm, 10 W, trans to cis) for 1 min–5 min, absorbance was recorded with the Shimadzu UV/VIS spectrophotometer UV-2600 and graphed with the software Origin 2022b.

**Thermodynamic stability.** The nucleic acid sequences were dissolved in 1× PBS standard buffer and annealed then transferred to a T_m_ cuvette, and the air bubbles were drained and tightly closed for the experiment, and the melting temperature curve was determined by UV (365 nm, 10 W) or visible light for 10 min. The melting curve was obtained by measuring the UV absorption at 260 nm at gradient temperature (Shimadzu UV/Vis spectrophotometer UV-2600 and temperature controller programmed temperature rise of 1 °C/min), recording the device and differentially calculating the T_m_ value using the software Matlab program discrete function.

**Primer extension Assays.** Mix the template strand with molar amounts of azobenzene-modified nucleotides in 2 μL of 10×RNA Gerpol reaction solution (10 mM NaCl; 40 mM Tris, pH 7.8; 6 mM MgCl_2_; 2 mM spermidine and 10 mM DTT) solution, add 1 μL dNTP 95 °C annealing for 5 min, cool naturally to room temperature, perform parallel experiments, one set of visible light illumination, one set irradiated with an ultraviolet lamp (365 nm) for 5 min, followed by incubation under dark conditions for 10 min to fully bind the template strands and azODNs. After adding Taq DNA polymerase (50 U/μL) and primers, the reactants are gently vortexed and incubated at 37 °C for 20 min, 30 min and 40 min, respectively, to obtain the extension product. After the reaction is complete, add the same volume of loading buffer (90% deionized formamide, 25 mM EDTA, 0.02% bromophenol blue) and incubate at 90 °C for 5 min. Take 10 μL of sample and inject 1× of TBE buffer into a 20% denaturing polyacrylamide gel containing 7 M urea, 150 V for 2 h. DNA templates and product of DNA primer extension isolated on the gel analyzed by a gel imaging system. Efficiency of primer extension was determined by percentage of PCR products, by dividing the intensity of the band corresponding to full-length products by total intensity of primers and products in gel.

**Kinetic data determination.** Mix the template strand with a molar amount of Az9 in 2 μL of 10 × Taq buffer reaction solution (10 mM NaCl; 40 mM Tris, pH 7.8; 6 mM MgCl 2; 2 mM spermidine and 10 mM DTT) solution, add 1 μL dNTP 95 °C annealed for 5 min, naturally cooled to room temperature, then incubated under dark conditions, parallel experiments are performed, one group is irradiated with an ultraviolet lamp (365 nm) for 0 min to 60 min, one group is not irradiated with UV lamp, samples are taken every 10 min, Taq DNA polymerase (50 U/μL) and primers are added, the reactants are gently vortexed and incubated at 37 °C for 1 h to obtain an extension product, respectively. After the reaction is complete, add the same volume of loading buffer (90% deionized formamide, 25 mM EDTA, 0.02% bromophenol blue) and incubate at 90 °C for 5 min. Take 10 μL of sample and inject 1× of TBE buffer into a 20% denaturing polyacrylamide gel containing 7 M urea, 150 V for 2 h. DNA templates and primer products isolated on the gel are analyzed by a gel imaging system.

Quantify the percentage of DNA generated at time t, fit discrete value s using a nonlinear curve, exponential function y = y_0_ + Ae^R0×t^ to obtain kinetic data for elongation products in the presence of even azODNs and Taq polymerases, for light irradiation, where y is the proportion of PCR products generated at time t and R_0_ is the rate constant of PCR product formation. The R_0_ of the black line (without UV treatment) is 0.024 and the red line (UV treatment) is 0.0056. The elongation efficiency without UV treatment is almost 4.3 times that of UV treatment.

**Reversible photomodulation assays.** Mix the template strand with a molar amount of Az9 in 2 μL of 10 × Taq buffer reaction solution (10 mM NaCl; 40 mM Tris, pH 7.8; 6 mM MgCl 2; 2 mM spermidine and 10 mM DTT) solution, add 1 μL dNTP 95 °C annealed for 5 min, naturally cool to room temperature, then incubate the binding nucleic acid strands under dark conditions, perform parallel experiments, one group irradiates with UV (365 nm), one group does not irradiate with ultraviolet lamps, after adding Taq polymerase (50 U/μL) and primers, gently vortex the reactants and incubate at 37 °C, samples are taken every 5 min (UV lamp groups irradiated for 2 min after 12 min and 26 min) to obtain the extension product, respectively. After the reaction is complete, add the same volume of loading buffer (90% deionized formamide, 25 mM EDTA, 0.02% bromophenol blue) and incubate at 90 °C for 5 min. Take 10 μL of sample and inject 1× of TBE buffer into a 20% denaturing polyacrylamide gel containing 7 M urea, 150 V for 2 h. DNA templates and primer products isolated on the gel are analyzed by a gel imaging system.

## Figures and Tables

**Figure 1 molecules-27-08992-f001:**
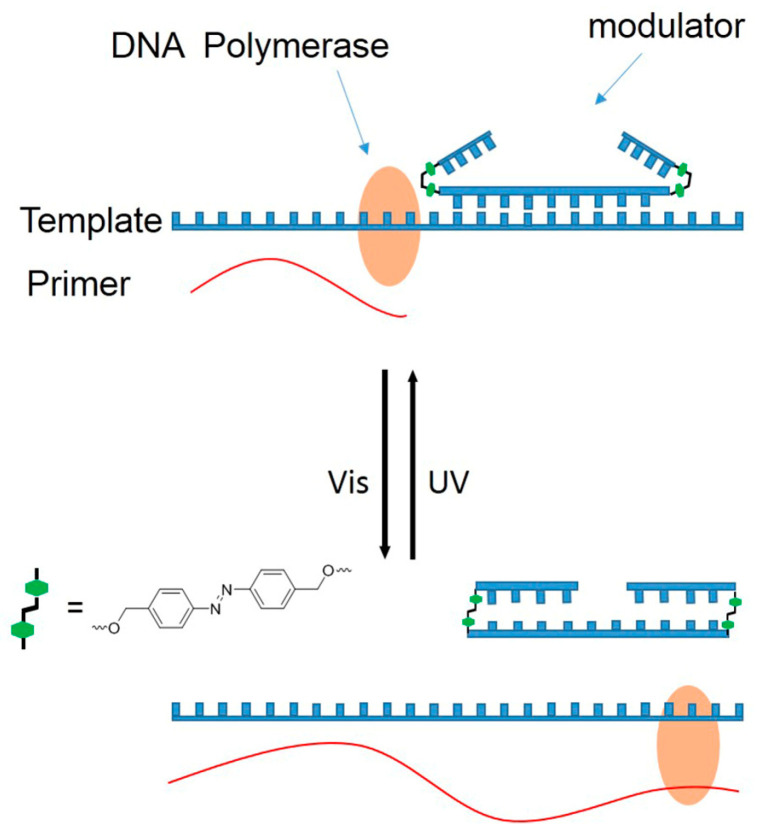
Strategy for photoregulating a primer extension reaction using azobenzene-linked dumbbell asODNs. UV irradiation can induce azODNs binding to the template, hindering primer extension and extending normally after visible irradiation.

**Figure 2 molecules-27-08992-f002:**
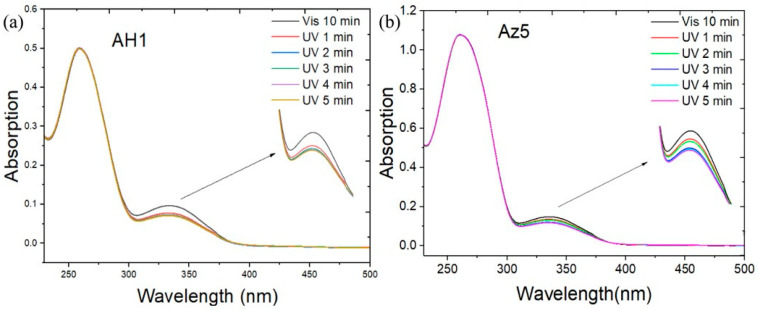
Photoisomerization of azobenzene-modified DNAs with UV or visible irradiation. The DNA sequences were (**a**) AH1: 5′-AAAGazoCTTT-3′ and (**b**) Az5: 5′-CGTTGazoCAACGTTTCGGACCGTATazoATACG-3′, respectively. The samples were irradiated for various times (1 min, 2 min, 3 min, 4 min and 5 min) with UV light at 365 nm or for 10 min with visible light.

**Figure 3 molecules-27-08992-f003:**
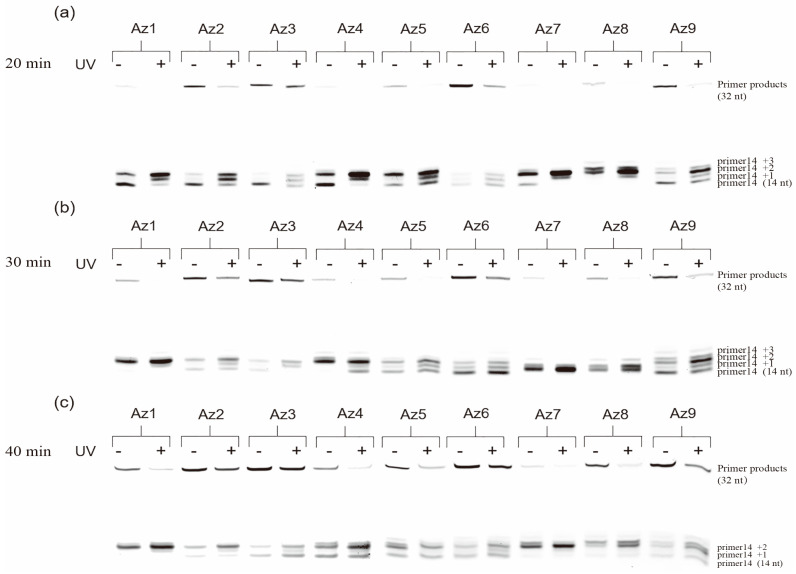
Denaturing PAGE (20%) of photomodulation of the primer extension reactions catalyzed by 1 U Taq DNA polymerase in the presence of 1 μM FAM-primer and Tem32 template, 1 μM azobenzene-linked asODNs and all four dNTPs (0.4 mM each) in 20 μL Taq buffer without (−) or with (+) UV irradiation at 37 °C. Polymerization reactions were incubated for (**a**) 20 min, (**b**) 30 min and (**c**) 40 min before terminating with EDTA.

**Figure 4 molecules-27-08992-f004:**
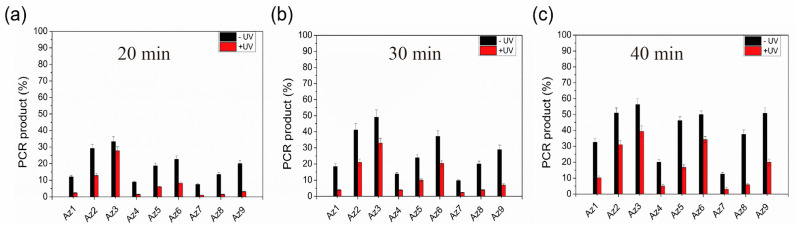
Quantitative analysis of photomodulation of the primer extension reactions catalyzed by 1 U Taq DNA polymerase in 20 μL Taq buffer containing 1 μM FAM-primer and Tem32 template, 1 μM azobenzene-linked asODNs and all four dNTPs (0.4 mM each). Polymerization reactions were incubated for (**a**) 20 min, (**b**) 30 min and (**c**) 40 min before terminating with EDTA. PCR product (%) refers to the percentage of full-length product of the primer extension reactions.

**Figure 5 molecules-27-08992-f005:**
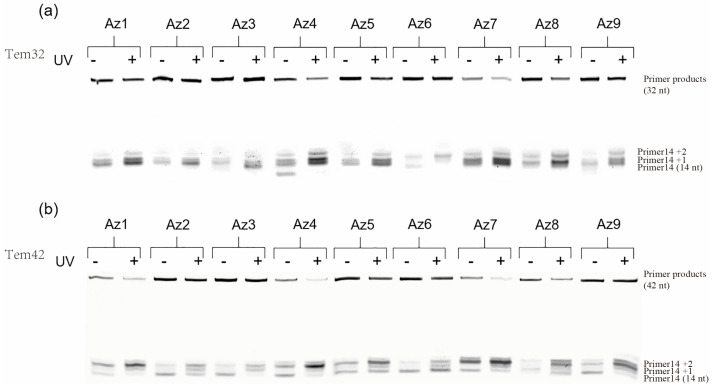
Denaturing PAGE (20%) of photomodulation of the primer extension reactions using (**a**) Tem32 or (**b**) Tem42 as template. Polymerization reactions without or with UV irradiation were performed for a long period of time (90 min) at 37 °C.

**Figure 6 molecules-27-08992-f006:**
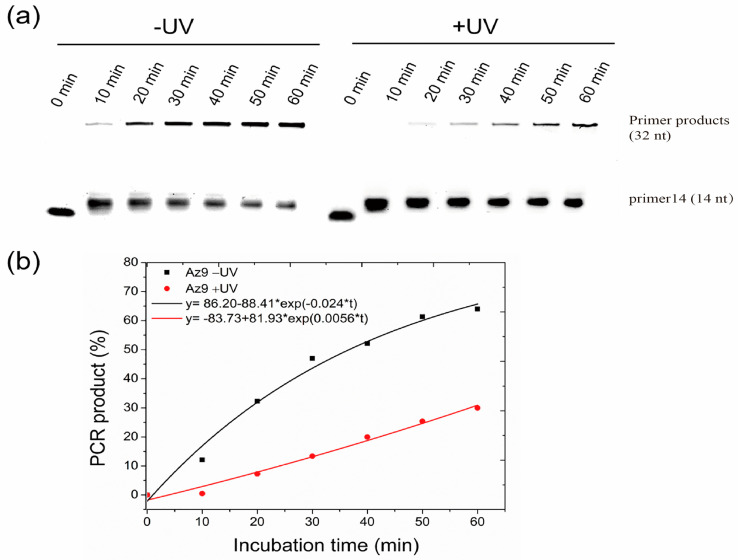
(**a**) Denaturing PAGE (20%) of photomodulaton of the primer extension by Taq polymerase in the presence of 1 μM FAM-primer and Tem32 template, 1 μM azobenzene-linked asODNs (Az9) and 1 U enzyme in 20 μL Taq buffer without (−) or with (+) UV irradiation at 37 °C with increasing incubation time. Aliquots of each incubated sample were removed at 0, 10, 20, 30, 40, 50 and 60 min. (**b**) Kinetics analysis for photoregulation of primer extension using Az9 in 20 μL Taq buffer containing 1 U enzyme at 37 °C with increasing incubation time. PCR Product (%) refers to the percentage of full-length product of the primer extension reactions.

**Figure 7 molecules-27-08992-f007:**
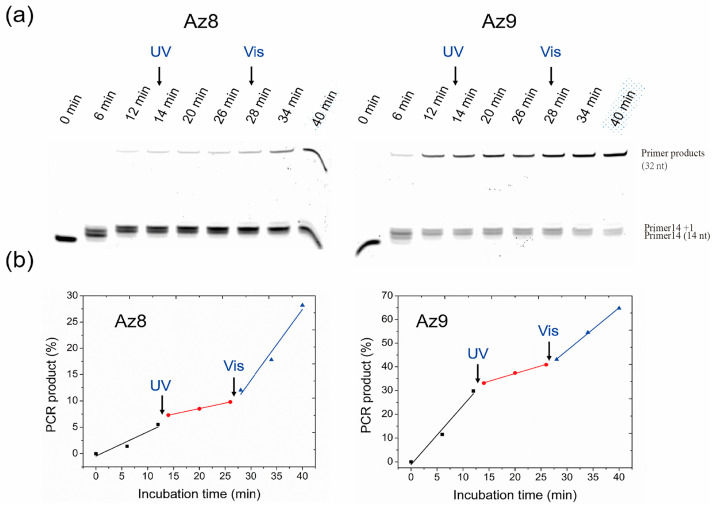
Photoswitching of primer extension using Az8 and Az9 by alternate photoirradiation between UV (2 min) and visible light (2 min). The reaction−mixed solution of 1 μM FAM-primer and Tem32 template, 1 μM azobenzene-linked asODNs (Az8 and Az9) was incubated in 1U Taq at 37 °C, and the aliquots were collected every 6 min except for irradiation time. (**a**) DNA products were separated by denaturing PAGE. (**b**) The relative yield of extension based on the conversion efficiencies of primer (10 min) was plotted. PCR Product (%) refers to the percentage of full-length product of the primer extension reactions.

**Table 1 molecules-27-08992-t001:** Sequences of oligonucleotides and azODNs, templates and primer used in this study.

azODNs Name	Base Sequence
A1	5′-AAAGTTTTCTTT-3′
A2	5′-AAAAGTTTTCTTTT-3′
A4	5′-AATAGTTTTCTATT-3′
AH1	5′-AAAGazoCTTT-3′
AH2	5′-AAAAGazoCTTTT-3′
AH3	5′-AATAGazoCTATT-3′
Az1	5′-CGTTazoAACGTTTCGGACCGTAazoTACG-3′
Az2	5′-ACGTTazoAACGTTTCGGACCGTAazoTACGG-3′
Az3	5′-AACGTTazoAACGTTTCGGACCGTAazoTACGGT-3′
Az4	5′-GTTGazoCAACGTTTCGGACCGTATazoATAC-3′
Az5	5′-CGTTGazoCAACGTTTCGGACCGTATazoATACG-3′
Az6	5′-ACGTTGazoCAACGTTTCGGACCGTATazoATACGG-3′
Az7	5′-TTGGazoCCAACGTTTCGGACCGTATTazoAATA-3′
Az8	5′-GTTGGazoCCAACGTTTCGGACCGTATTazoAATAC-3′
Az9	5′-CGTTGGazoCCAACGTTTCGGACCGTATTazoAATACG-3′
Primer14	5′-FAM-AATACGACTCACCC-3′
Tem32	3′-TTATGCTGAGTGGGTTGCAAAGCCTGGCATAA-5′
Tem42	3′-TTATGCTGAGTGGGTTGCAAAGCCTGGCATAATTTTTTTTTT-5′

**Table 2 molecules-27-08992-t002:** The melting temperatures of natural hairpin nucleotides, azobenzene-linked hairpin oligonucleotides and azobenzene-linked dumbbell oligonucleotides.

ODNs Name	T_m_ (Vis, °C)	T_m_ (UV, °C)
A1	35.2 ± 0.1	-
A2	33.4 ± 0.0	-
A3	33.5 ± 2.1	-
AH1	53.0 ± 0.2	38.9 ± 1.0
AH2	58.6 ± 1.4	40.0 ± 3.6
AH3	53.6 ± 0.7	40.4 ± 1.0
Az1	62.5 ± 1.0	50.6 ± 0.4
Az2	71.5 ± 0.0	61.6 ± 1.6
Az3	73.5 ± 0.5	59.6 ± 0.4
Az4	68.4 ± 0.8	45.5 ± 0.4
Az5	69.6 ± 0.0	53.3 ± 0.0
Az6	75.5 ± 0.8	60.5 ± 0.6
Az7	50.9 ± 0.0	35.4 ± 0.7
Az8	70.6 ± 1.2	57.8 ± 1.3
Az9	71.7 ± 0.5	57.0 ± 0.1

## Data Availability

The experimental data provided in this work are available in articles and Appendix A.

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
