# Peer review of "Reversible On-Off Photoswitching of DNA Replication Using a Dumbbell Oligodeoxynucleotide"

_molecules, 2022, doi:10.3390/molecules27248992_

Round 1
Reviewer 1 Report
This is a very nice piece of work by Wu and co-workers, which shows a novel approach for the photo-regulation of DNA polymerase extension using azobenzene-embedded dumbbell linkers. Through systematical investigations on the effects of these modified ODN linkers, the authors demonstrate the azobenzene-based photo-responsive systems can be utilized to manipulate the enzymatic activity of the Taq DNA polymerase and further modulate the synthesis of DNA strands by light in a reversible way. Overall, this work is well-designed, and experiments are carefully performed. Thus, I recommend publication of this manuscript after some minor revisions.
Some specific comments:
1) In this work, thermodynamic stability of the DNA hairpins decreased sharply from trans-to-cis transformation, as shown in Table 2. But another article showed that the trans-form of the azobenzene linker caused opening of the hairpin structure (Tetrahedron Letters. 2008, 49(34): 5087-5089). The contradiction should be discussed, which would help readers to understand the different rationales.
2) Why does the gel of the dumbbell nucleic acid Az1 in Figure 3a appear differently than in Figures 3b and 3c? In the absence of UV treatment, two bands were showed in Figure 3a, whereas only one band was showed in Figure 3b and 3c.
3) Line 27: “Azobenzene has been shown to be a reversible manipulated light-response molecule [7]”, The citation of reference 7 does not match the meaning of the above sentence.
4) Line 81: “azODN” is not defined.
5) Line 136-137: “hairpining” should be “hairpin”. Please check it through the whole text.
6) Line 205: “the two binding arms of the asODN are opened and stably bound to the template under the synergistic effect of free nucleotides…” Is it asODN or azODN in this sentence?
7) Line 346: “The mixture was stirred under N2 and ……”, Here the number “2” should appear as a subscript.
8) Line 356: “All nucleic acid sequences are connected to the solid-phase CPG from the 3' end to the 5' end using the DNA/RNA solid-phase synthesizer by de-DMT…” What is “de-DMT”?
9) Line 404: The product of DNA primer extension is not an RNA product.
10)Line 421: the linear equation "[S] = A + kt" does not correspond to Figure 6.
11)Line 432 and 436, please check the use of ºC symbol.
Author Response
Dear Editor
Thanks for your kind help for our manuscript entitled: “Reversible on-off photoswitching of DNA replication using a dumbbell oligodeoxynucleotide” (Manuscript ID: molecules-2050250). We answer all the questions raised by the reviewers carefully. We have carefully modified the article format and other issues according to your requirements. The specific details are as follows.
Some specific comments
1) In this workthermodynamic stability of the DNAhairpins decreased sharply from trans-to-cis transformation. as shown in Table 2.But another article showed that the trans-form of the azobenzene linker caused opening of the hairpin structure(Tetrahedron Letters. 200849(34):5087-5089)The contradiction should be discussed.which would help readers to understand the different rationales
Answer: Thanks for the suggestion. The azobenzene linkers used in our manuscript is 4,4’-bis(hydroxyethyl)-azobenzene (Figure 1), and (Tetrahedron Letters. 200849(34):5087-5089) the azobenzene used is 3,3’-bis(hydroxyethyl)-azobenzene (Figure 2) in this article. These are two completely different structural types of azo benzene analogues, structure determines properties, our experimental results show that trans-azo benzene oligonucleotides present hairpin structure (
https://doi.org/10.1039/C4MD00378K; https://doi.org/10.1021/acs.bioconjchem.5b00125). This may be related to the position and length of chemical substitution, and exploring the difference in the performance of different types of azobenzene in nucleic acids requires further simulation and confirmation.
2)Why does the gel of the dumbbell nucleic acid Az1in Figure 3a appear differently than inFiqures 3b and 3c? In the absence of UV treatmenttwo bands were showed in Fiqure 3a whereas only one band was showed in Figure 3b and 3c
Answer: Thanks for the suggestion. The reason there are multiple bands is the influence of azODNs. Normally, primers extend to the end until obstructed by azODNs. Each additional nucleotide in the reverse strand of azODNs results in one more base in the primer extension length (Primer 14 +1,+2,+3 and +4), because the number of bases in the complementary pairing of the primer with the template strand is one less.
3)Line 27:“Azobenzene has been shown to be a reversible manipulated light-response
molecule [7]",The citation of reference 7 does not match the meaning of the above sentence
Answer: Thanks for your comment. We have deleted previous Ref 7 and inserted one new matching references (Thevarpadam, J.; Bessi, I.; Binas, O.; Goncalves, D. P.; Slavov, C.; Jonker, H. R.; Richter, C.; Wachtveitl, J.; Schwalbe, H.; Heckel, A., Photoresponsive Formation of an Intermolecular Minimal G-Quadruplex Motif. Angew Chem Int Ed Engl 2016, 55 (8), 2738-42) in the text.
- Line 81:“azODN”is not defined
Answer: Thanks for your comment. We have defined "asODN" and his full name is antisense oligonucleotide. In addition, we put the interpretation of azODN on line 81 and removed the abbreviated explanation in line 89 to 91.
- Line 136-137:“hairpining”should be“hairpin”.Please check it through the whole text.
Answer: Thanks for your comment. We changed "hairpinning" to "hairpin".
6) Line 205:“the two binding arms of the asODN are opened and stably bound to the templateunder the synergistic effect of free nucleotides….”Is it asODN or azODN in this sentence?
Answer: Thanks for your comment. In this paper, asODNs usually refer to natural antisense nucleotides, and azODNs refer to dumbbell-type antisense nucleotides that connect two azobenzenes. This should refer to “azODNs”, which we have removed and replaced with “azODNs”.
7) Line 346:“The mixture was stirred under N2 and…”Here the number“2”should appear
as a subscript.
Answer: Thanks for your comment. We have changed “N2” to “N2”. We modify “de-” to “removing”.
8)Line 356:“All nucleic acid sequences are connected to the solid-phase CPG from the 3end
to the 5'end using the DNA/RNAsolid-phase synthesizer by de-DMT…."What is“de-DMT"?
Answer: Thanks for your suggestion. What we're trying to say is to take off DMT. We have changed "...... using the DNA/RNA solid-phase synthesizer by de-DMT......" into "...... the DNA/RNA solid-phase synthesizer by removing DMT.....".
9) Line 404: The product of DNA primer extension is not an RNA product.
Answer: Thanks for your comment. We have changed “the RNA products” into “primer extension product”.
10)Line 421:the linear equation"[S]=A+kt" does not correspond to Fiqure6
Answer: Thanks for your comment. We rewrote this related paragraph.(Quantify the percentage of DNA generated at time t, fit discrete values using a nonlinear curve, exponential function y = y0+AeR0*t to obtain kinetic data for elongation products in the presence of even azODNs and Taq polymerases, for light irradiation, where y is the proportion of PCR products generated at time t and R0 is the rate constant of PCR product formation. The R0 of the black line (without UV treatment) is 0.024 and the red line (UV treatment) is 0.0056. The elongation efficiency without UV treatment is almost 4.3 times that of UV treatment.)
- Line 432 and 436. please check the use ofC svmbol
Answer: Thanks for your suggestion. We checked the units of degrees Celsius for the whole text, and unified them in oC
We hope to meet the requirements of the journal this time.
With best regards,
Yours sincerely,
Li Wu
Reviewer 2 Report
In this study the authors position UV-modifiable analogs in primers that can then be inter-converted using exposure to light. What is absent from this paper in the Introduction is why is this beneficial? SO if primer extension with Taq is done using the same length DNA, what do the products look like on a gel? Does the incorporation of the modified primers provide a significant benefit? If yes insight into how much of a benefit would be ideal.
In this study the authors use primer extension but the gels are unlabeled and under-exposed so it is impossible to know what is happening and thus make any conclusions.
Figure 3 is impossible to understand as bands are unlabeled.
Figure 4 is difficult to comprehend as it shows quantification of primer extension reactions. Are these the same reactions as in Fig 3? If yes, which bands are the product?
Figure 5 is unlabeled so it is impossible to know products from substrate versus wells
Figure 6a bands are unlabeled. SO there are 2 possible products with the upper most band the wells? This is unclear
Figure 7a bands are unlabeled so it is impossible to know what is going on.
All bands in all gels need to be labeled and explained carefully.
Figure legends for all figures are very poor and brief. They lack the detail required to understand the figures.
Author Response
Dear Editor
Thanks for your kind help for our manuscript entitled: “Reversible on-off photoswitching of DNA replication using a dumbbell oligodeoxynucleotide” (Manuscript ID: molecules-2050250). We answer all the questions raised by the reviewers carefully. We have carefully modified the article format and other issues according to your requirements. The specific details are as follows.
In this study the authors position UV-modifiable analogs in primers that can then be inter-converted using exposure to light. What is absent from this paper in the Introduction is why is this beneficial? SO if primer extension with Taq is done using the same length DNA, what do the products look like on a gel? Does the incorporation of the modified primers provide a significant benefit? If yes insight into how much of a benefit would be ideal.
Answer: Thanks for the suggestion. We add the meaning of this new strategy after "......hoping to manipulate primer extension artificially, quickly and efficiently." (We explored another strategy to regulate gene expression, combining azODNs to template strands (DNA single strands), which affect the extension of enzymes through the trans- and cis- structures of azobenzene, controlling gene expression. This method is very flexible, and can synthesize fragment-length azODNs in vitro for the target gene sequence you want to study, and even develop it for use on plasmids in cells. This will lay the foundation for us to explore the mechanisms of action between subs.)
If only primer extension is used, his strip and template chain are in the same position after SYBR staining. So here we use FAM-labeled primers, and primer extension bands can be stained without SYBR dye and can still be observed in the gel imaging system, excluding the interference of other nucleic acid strands (including template strands). The primer extension product can be clearly obtained.<Bioconjugate Chem. 2017, 28, 8, 2125-2134>
In this study the authors use primer extension but the gels are unlabeled and under-exposed so it is impossible to know what is happening and thus make any conclusions.
Figure 3 is impossible to understand as bands are unlabeled.
Answer: Thanks for the suggestion. We standardize Figure 3 and replace the original Figure 3
Figure 4 is difficult to comprehend as it shows quantification of primer extension reactions. these the same reactions as in Fig 3? If yes, which bands are the product?
Answer: Thanks for the suggestion. Yes, these are the same reactions as in Fig 3. We have labeled Figure 4a, 4b and 4c with 20 min, 30 min and 40 min to help make it easier to compare with Figure 3.
Figure 5 is unlabeled so it is impossible to know products from substrate versus wells
Answer: Thanks for the suggestion. We annotated the primers, primer products and template classifications in Figure 5, replacing the original Figure 5. Below is primer14, and above is an extension product of primer, Figure 5a is a control experiment on template Tem32 and Figure 5b is a control experiment on template Tem32.
Figure 6a bands are unlabeled. SO there are 2 possible products with the upper most band the wells? This is unclear
Answer: Thanks for the suggestion. We annotated the primers and primer products in Figure 6a, replacing Figure 6a. Below is primer14, and above is an extension product of primer.
Figure 7a bands are unlabeled so it is impossible to know what is going on.
Answer: Thanks for the suggestion. We annotated the primers and primer products in Figure 7a, replacing the original Figure 7a. Below is primer14, and above is an extension product of primer.
We hope to meet the requirements of the journal this time.
With best regards,
Yours sincerely,
Li Wu

Reviewer 3 Report
The importance of controlling DNA replication processes is still one of the main topics of molecular and cellular biology. To regulate of DNA chain elongation under UV irradiation, the authors proposed a new strategy for creating chimeric structures based on the incorporation of azobenzene into oligonucleotide chains. In addition, the possibility of reversibility of the process of photoregulation of primer elongation under alternate irradiation with UV and visible light was studied.
However, despite the obvious relevance of the problem and the importance of the obtained results, the article requires extensive revision and correction of the English language. In the presented form, the article cannot be published in the journal.
The article “Reversible on-off photoswitching of DNA replication using a dumbbell oligodeoxynucleotide” by Yu Wang et al. may be published in the “Molecules” journal after serious revision.
MAJOR REVISION
1. Some conclusions of the authors are unfounded. Authors write: «We have shown that asODNs can be inactivated in an efficient manner tominimally silence gene expression by using UV light, 365nm, and that reactivation by visible light can return asODNs to efficient gene silencing substrates.» How did the authors test the effect of gene suppression? This is a very serious conclusion and experimental confirmation is needed. It is necessary to present the results of studies or not to mention such effects so explicitly.
2. In supplementary materials, the authors present the NMR spectra of the studied samples. An analysis of these data should be included in the Results chapter. The describe of NMR method and obtained results should be added In supplementary materials.
3. For a clearer presentation of the material, Table S1 must be included in the main text.
4. Please provide on Figure 7 a better electrophoregram. Also the DNA molecular weight marker is not presented (Figure 3, 5-7).
5. Some phrases of the authors are poorly written, which makes it difficult to understand the text. For example (Line 131-135):
To study the effect of temperature on the structure of azobenzene nucleotide, by judging the Tm value of the melting chain temperature, the thermodynamic stability of the oligonucleotide reflects the binding strength of the complementary double strand, and by detecting the change of the dissolution chain temperature of each oligonucleotide sequence before and after illumination, it can be speculated whether the photoisomerization of azobenzene derivatives affect the thermodynamic stability.
6. Statistical errors don’t present in the Table 2.
7. Chapter 2.5. Line 230-232
«the effect of azODNs on reversible photomodulation of primer extension was investigated, alternating light irradiation between ultraviolet and visible light»
For investigating the effect of azODN on reversible photomodulation of primer extension under alternating UV and visible light irradiation, Az8 and Az9 were selected. It is necessary to explain in the text why these azODNs were chosen. For a better understanding of the results of the work, please add a table indicating Tm for all sequences.
8. «We consider this to be an improvement over the traditional cage approach [24]. Because it has the potential to activate or close several times during treatment, whereas the traditional light cage releases its full load once activated and is not affected by light afterward.»
It is necessary to describe in more detail the traditional approaches and clearly specify the main advantages of the new approach proposed by the authors.
MINOR REVISION
1. Line 69-70.
«at temperatures below oligonucleotide Tm». It must be clarified that Tm is the melting temperature.
2. Line 73.
«Therefore, the possibility of including azo in genes by PCR is limited»
«azo» – what is it?
3. The authors use several terms: Azobenzene (line 53), azo (line 73), BAzobenzene (line 58). Do these terms describe the same compound?
4. Line 75, 88
The text contains an inaccurate interpretation of asODNs and azODNs. The authors write "antisense oligonucleotides (asODNs)" and "Azobenzene antisense oligonucleotides (azODNs)". The literature data: "asODNs - antisense oligodeoxynucleotides" and also authors write «oligodeoxynucleotide» in the the title of the article.
5. Line 90-91
«and each hairpin contains inhibitory sensory chains 90 of 4-mer, 5-mer, and 6-mer lengths at both ends of these ODNs».
Please bring the abbreviations in the text in accordance with the data presented in the tables. In addition, Table 1 shows the sequence of Primer14, but this primer is not mentioned in the main text.
6. Figure 2.
Az5 5’-CGTTGazoCAACGTTTCGGACCGTATazoATACG-3. Not marked 3' end
Az5 5’-CGTTGazoCAACGTTTCGGACCGTATazoATACG-3' (Table 1)
7. Line 144-145
«18.6 oC Celsius and the lowest 144 14.1 oC Celsius». No need to specify Celsius
8. Line 162
14-mer is Primer14? 32-mer DNA template is Tem32. It is necessary to be explained in the text.
9. Line 173, 169
14 nt, 32 nt
Explain, please, in the text and in the legend of Fig. 3. these parameters
10. Table 1 and Figure 5
In the Legend for Fig.5 authors write «32mer-template (Tem32)» and «42-mer-template (Tem42)». However, in the text of the article, the authors use the concept of 32mer and 42mer (line 214), and in Table 1 - Tem32 and Tem42. Please enter explanations in the text itself. The authors use the term primer14 only in Table 1; there is 14nt in the text (line 169) is Primer14? Please explain in the text of the article.
11. Figure 3, Legend
«buffer without or with UV irradiation» must be replaced by «buffer without or with UV irradiation» «buffer without (-) or with (+) UV irradiation»
12. Line 216
«function of enough time (90min) (Figure 6).». Probably the authors are discussing the result presented in Figure 5, and not in Figure 6.
13. From the point of view of the English language, it seems to me that the article requires significant revision.
Author Response
Dear Editor
Thanks for your kind help for our manuscript entitled: “Reversible on-off photoswitching of DNA replication using a dumbbell oligodeoxynucleotide” (Manuscript ID: molecules-2050250). We answer all the questions raised by the reviewers carefully. We have carefully modified the article format and other issues according to your requirements. The specific details are as follows.
- Some conclusions of the authors are unfounded. Authors write: «We have shown that asODNs can be inactivated in an efficient manner tominimally silence gene expression by using UV light, 365nm, and that reactivation by visible light can return asODNs to efficient gene silencing substrates.» How did the authors test the effect of gene suppression? This is a very serious conclusion and experimental confirmation is needed. It is necessary to present the results of studies or not to mention such effects so explicitly.
Answer: Thanks for the suggestion. In our previous work (Photoregulating RNA digestion using azobenzene linked dumbbell antisense oligodeoxynucleotides. Bioconjug Chem 2015, 26 (6), 1070-9), Using azobenzene linked dumbbell antisense oligodeoxynucleotides to light-controlled binding to RNA, RNA degradation by recruiting RNase H (Figure 1).
Figure 1
After thinking, we decided to change this sentence «We have shown that asODNs can be inactivated in an efficient manner tominimally silence gene expression by using UV light, 365nm, and that reactivation by visible light can return asODNs to efficient gene silencing substrates.» to "Experimental results show that we can use azODNs to artificially experiment with visible light regulation to turn on primer extension, and use ultraviolet control to inhibit primer extension"
- In supplementary materials, the authors present the NMR spectra of the studied samples. An analysis of these data should be included in the Results chapter. The describe of NMR method and obtained results should be added In supplementary materials.
Answer: Thanks for the suggestion. NMR data is used to characterize whether the azo benzene we synthesized is pure and correct. There is no point in further analysis of the data, because our focus is on nucleic acid regulation, so the NMR data only needs to help us identify the chemical samples we need. In chemical synthesis, the NMR data results are usually placed at the end of the experimental methodological steps, and of course, the NMR diagram needs to be placed in the supporting information. We put the steps and results of azobenzene synthesis in materials and methods to help readers understand our working ideas more conveniently.
- For a clearer presentation of the material, Table S1 must be included in the main text.
Answer: Thanks for the suggestion. We transferred the names and sequences of Table S1 from all to Table 1.
- Please provide on Figure 7 a better electrophoregram. Also the DNA molecular weight marker is not presented (Figure 3, 5-7).
Answer: Thanks for the suggestion. We relabeled the primers and primer products in Figures 3, 5, 6, and 7, and replaced the previous images.
- Some phrases of the authors are poorly written, which makes it difficult to understand the text. For example (Line 131-135):
To study the effect of temperature on the structure of azobenzene nucleotide, by judging the Tm value of the melting chain temperature, the thermodynamic stability of the oligonucleotide reflects the binding strength of the complementary double strand, and by detecting the change of the dissolution chain temperature of each oligonucleotide sequence before and after illumination, it can be speculated whether the photoisomerization of azobenzene derivatives affect the thermodynamic stability.
Answer: Thanks for your suggestion. We deleted the lengthy text and replaced it with the short text "The alteration of thermodynamic stability caused by isomerization of azobenzene is a necessary prerequisite for azODNs to inhibit DNA elongation"
- Statistical errors don’t present in the Table 2.
Answer: Thanks for your suggestion. The STDEV function in EXCEL was used to estimate the standard deviation, and the values were added in Table 2. The uniform number of digits is retained to one decimal place.
- Chapter 2.5. Line 230-232
«the effect of azODNs on reversible photomodulation of primer extension was investigated, alternating light irradiation between ultraviolet and visible light»
For investigating the effect of azODN on reversible photomodulation of primer extension under alternating UV and visible light irradiation, Az8 and Az9 were selected. It is necessary to explain in the text why these azODNs were chosen. For a better understanding of the results of the work, please add a table indicating Tm for all sequences.
Answer: Thanks for your suggestion. When Az8 and Az9 are applied to primer extension, the comparison between 20 minutes and 40 minutes is more obvious than other azODNs, which can help us analyze the influence of hairpin base numbers at both ends of dumbbell oligonucleotides on primer extension.
We explain why we chose Az8 and Az9 at the end of the first paragraph under Level 2.5. "In the Analysis of primer-extension reactions with azODNs, it was found that Az8 and Az9 had different primer extension regulation results at 20 min and 40 min, and Az8 and Az9 had the same complementary base number for the template chain, and the number of hairpins at both ends was different by 1 base, so Az8 and Az9 were selected for controllable primer extension experiments."
We added the Tm values of the measured azODNs (Az1-Az9) to Table 2.
- «We consider this to be an improvement over the traditional cage approach [24]. Because it has the potential to activate or close several times during treatment, whereas the traditional light cage releases its full load once activated and is not affected by light afterward.»
It is necessary to describe in more detail the traditional approaches and clearly specify the main advantages of the new approach proposed by the authors.
Answer: Thanks for your suggestion. We have added more advantages of azobenzene later. (Because the photocage group needs to be synthesized with the base of the nucleotide monomer, once the NPOM can left by ultraviolet light, it cannot be restored to the state of modified nucleotide monomer, and NPOM is a reaction product after illumination, which may also have an effect on organisms. In addition to the advantage that azo-benzene can be switched on and off repeatedly, chemical synthesis of azobenzene is easier than traditional photocage’s group NPOM.)
MINOR REVISION
- Line 69-70.
«at temperatures below oligonucleotide Tm». It must be clarified that Tm is the melting temperature.
Answer: Thanks for your suggestion. We explain what Tm means (melting temperature).
- Line 73.
«Therefore, the possibility of including azo in genes by PCR is limited»
«azo» – what is it?
Answer: Thanks for your suggestion. The “azo” is “azobenzene-modified nucleotides”, and we modified the original text to replace azo with Azobenzene-modified nucleotides.
3The authors use several terms: Azobenzene (line 53), azo (line 73), BAzobenzene (line 58). Do these terms describe the same compound?
Answer: Thanks for your suggestion. Azobenzene (line 53), azo (line 73), BAzobenzene (line 58). They are the same compound in this article, azo is a generic short for azobenzene. But BAzobenzene was mistyped and added a 'B', and we modified the original text.
- Line 75, 88
The text contains an inaccurate interpretation of asODNs and azODNs. The authors write "antisense oligonucleotides (asODNs)" and "Azobenzene antisense oligonucleotides (azODNs)". The literature data: "asODNs - antisense oligodeoxynucleotides" and also authors write «oligodeoxynucleotide» in the the title of the article.
Answer: Thanks for your suggestion. We examined the full text redefining asODNs as antisense oligodeoxynucleotides and azODNs as azobenzene oligodeoxynucleotides
- Line 90-91
«and each hairpin contains inhibitory sensory chains 90 of 4-mer, 5-mer, and 6-mer lengths at both ends of these ODNs».
Please bring the abbreviations in the text in accordance with the data presented in the tables. In addition, Table 1 shows the sequence of Primer14, but this primer is not mentioned in the main text.
Answer: Thanks for your suggestion. We will, "... to be 16 nt, 18 nt, and 20 nt long..." and "... to 4-mer, 5-mer, and 6-mer...", all described in more detail according to Table 1, replaced by "... are designed to be 16 nt (Az1, Az2 and Az3), 18 nt (Az4, Az5 and Az6), and 20 nt (Az7, Az8 and Az9) long, Each sequence contains two azobenzenelinked hairpin structures, and each hairpin contains inhibitory sensory chains of 4-mer (Az1, Az4 and Az5), 5-mer (Az2, Az5 and Az8), and 6-mer (Az3, Az6 and Az9) lengths...", and The primer DNA (14mer) is labeled a fluorescein moiety (FAM)..." The primer DNA (14mer) refers to Primer14, which is changed to Primer14 in the text.
- Figure 2.
Az5 5’-CGTTGazoCAACGTTTCGGACCGTATazoATACG-3. Not marked 3' end
Az5 5’-CGTTGazoCAACGTTTCGGACCGTATazoATACG-3' (Table 1)
Answer: Thanks for your suggestion. We have corrected it in Figure 2.
- Line 144-145
«18.6 oC Celsius and the lowest 144 14.1 oC Celsius». No need to specify Celsius
Answer: Thanks for your suggestion. Due to our mistakes, in this sentence '... reach up to 18.6 oC Celsius and the lowest 14.1 oC Celsius...', “Celsius” is redundant, we delete it.
- Line 162
14-mer is Primer14? 32-mer DNA template is Tem32. It is necessary to be explained in the text.
Answer: Thanks for your suggestion. Your guesses are correct, and we have corrected them in the text. We changed the 14-mer to the Primer14.
- Line 173, 169
14 nt, 32 nt
Explain, please, in the text and in the legend of Fig. 3. these parameters
Answer: Thanks for your suggestion. We annotated primer 14 and primer extension products in Figure 3 and replaced the original figure.
- Table 1 and Figure 5
In the Legend for Fig.5 authors write «32mer-template (Tem32)» and «42-mer-template (Tem42)». However, in the text of the article, the authors use the concept of 32mer and 42mer (line 214), and in Table 1 - Tem32 and Tem42. Please enter explanations in the text itself. The authors use the term primer14 only in Table 1; there is 14nt in the text (line 169) is Primer14? Please explain in the text of the article.
Answer: Thanks for your suggestion. We checked the full text, and decided to replace 32-mer and 32mer with Tem32 in Table 1, and 42-mer with Tem42.
‘14nt’ in the text (line 169) is ‘Primer14’, after consideration, we decided to remove this 14nt because the phrase could be better understood.
- Figure 3, Legend
«buffer without or with UV irradiation» must be replaced by «buffer without or with UV irradiation» «buffer without (-) or with (+) UV irradiation»
Answer: Thanks for your suggestion. We modified the same issue that occurred in Figure 3 and Figure 6.
- Line 216
«function of enough time (90min) (Figure 6).». Probably the authors are discussing the result presented in Figure 5, and not in Figure 6.
Answer: Thanks for your suggestion. Yes, here we are talking about Figure 5 not Figure 6, which has been modified in the original text.
- From the point of view of the English language, it seems to me that the article requires significant revision
Answer: Thanks for your suggestion. After revisions, we carefully check the entire manuscript for grammatical errors and correct them.
We hope to meet the requirements of the journal this time.
With best regards,
Yours sincerely,
Li Wu

Round 2
Reviewer 2 Report
The authors had done a poor job improving the figures and legends. They still need significant improvement. In some Figures including gels, bands are labeled in one panel but not others - see Fig 3, 5, 6 and 7 as examples. All gels need to have all bands labeled.
Changes still to be made:
Fig 2. The peaks where changes are occurring are very difficult to discern. I suggest making an inset of only the 335nm peak for both panels a and b so that the changes can be seen. As it stands now the graph is optimized for the peak at 260 nm which does not change.
Fig 4. Is the analysis of the gels in Fig 3? If yes state this in the legend. If it is analysis of the gels in Fig 3 what does %Product mean? This has to be explained. Is it % of lane? input substrate? How is it calculated?
Fig 5. I have no idea what the Fig title means "Control of azODNs for different lengths of stencil extension." is incomprehensible and the legend is poor.
Fig 6. What is %Product? This has to be explained. Is it % of lane? input substrate? How is it calculated?
Fig 7. What is %Product? This has to be explained. Is it % of lane? input substrate? How is it calculated?
Author Response
Dear Editor
Thanks for your kind help for our manuscript entitled: “Reversible on-off photoswitching of DNA replication using a dumbbell oligodeoxynucleotide ” (Manuscript ID: molecules-2050250). We answered all the questions raised by the reviewers carefully. We have carefully modified the article format and other issues according to your requirements; we hope to meet the requirements of the journal this time. The specific details are as follows.
Fig 2. The peaks where changes are occurring are very difficult to discern. I suggest making an inset of only the 335nm peak for both panels a and b so that the changes can be seen. As it stands now the graph is optimized for the peak at 260 nm which does not change.
Answer: Thank you for the suggestion. We zoomed in on the absorption spectra around 335 nm 335 nm and inserted the magnified image in Figure 2 (a) and (b). The peak at 260 nm in the UV spectra of azobenzene linked ODNs slightly increase with UV irradiation. The increase is more obvious for azobenzene linked hairpins compared with azobenzene linked dumbbell ODNs.
Fig 4. Is the analysis of the gels in Fig 3? If yes state this in the legend. If it is analysis of the gels in Fig 3 what does %Product mean? This has to be explained. Is it % of lane? input substrate? How is it calculated?
Answer: Thank you for the suggestion. Yes, Figure 4 is an analysis of Figure 3. We added this in line 171 of the text "..... ... and quantified using denaturing polyacrylamide gel electrophoresis (PAGE) (Figure 3 and Figure 4) .......". The same experimental system as in Figure 3 is added to legend of Figure 4.
“%Product” means the yield of the percentage of the full-length product. We added labels to the tail of the figure 4 and line 198. Yes, it is a percentage of the lane, and refers to the percentage of primer14 extend full-length products.
We don't have an additional input substrate. We used the computer software Image Lab to calculate the percentage of products, and then used the software origin2021b to make the graph. We set the total intensity of all bands in a single lane to 100% (since they are rimer14 and Primer14 extension products with FAM), and the ratio of the intensity of the primer extension full-length product to the total intensity of this lane, multiplied by 100% is the percentage of product. We enter this percentage into the software origin2021b to get Figure 4. We focus on the product of the full-length extension.
For the calculation method of PCR product percentage, we added it to Materials and Methods.
Fig 5. I have no idea what the Fig title means "Control of azODNs for different lengths of stencil extension." is incomprehensible and the legend is poor.
Answer: Thank you for the suggestion. We modified the legend of Figure 5. <Denaturing PAGE (20%) of photomodulation of the primer extension reactions using (a) Tem32 or (b) Tem42 as template. Polymerization reactions without or with UV irradiation were performed for plenty of time (90 min) at 37 oC.>
Fig 6. What is %Product? This has to be explained. Is it % of lane? input substrate? How is it calculated?
Answer: Thank you for the suggestion. “%Product” means the yield of the percentage of the full-length product. We added labels to the tail of the figure 6. Yes, it is a percentage of the lane, and refers to the percentage of primer14 extend full-length products.
We don't have an additional input substrate. We used the computer software Image Lab to calculate the percentage of products, and then used the software origin2021b to make the graph. We set the total intensity of all bands in a single lane to 100% (since they are all derived from primer14 with the FAM mark), and the ratio of the intensity of the primer extension full-length product to the total intensity of this lane, multiplied by 100% is the percentage of product. We enter this percentage into the software origin2021b to get Figure 6. We focus on the product of the full-length extension.
Fig 7. What is %Product? This has to be explained. Is it % of lane? input substrate? How is it calculated?
Answer: Thank you for the suggestion. “%Product” means the yield of the percentage of the full-length product. We added labels to the tail of the figure7. Yes, it is a percentage of the lane, and refers to the percentage of primer14 extend full-length products.
We don't have an additional input substrate. We used the computer software Image Lab to calculate the percentage of products, and then used the software origin2021b to make the graph. We set the total intensity of all bands in a single lane to 100% (since they are all derived from primer14 with the FAM mark), and the ratio of the intensity of the primer extension full-length product to the total intensity of this lane, multiplied by 100% is the percentage of product. We enter this percentage into the software origin2021b to get Figure 7. We focus on the product of the full-length extension.
We hope to meet the requirements of the journal this time.
With best regards,
Yours sincerely,
Li Wu
Reviewer 3 Report
Dear authors, thank you for your comments.
The article can be accepted for publication in the “Molecules” journal.
Author Response
Thank you for the review, we have made some changes to the language again.